# Rethinking Attentions in Zero-Shot Real Image Editing

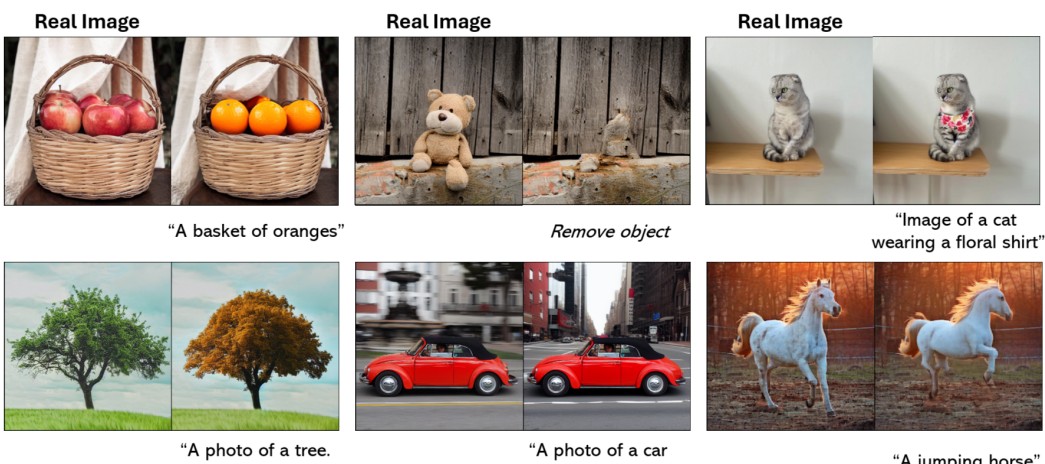

Figure 1: For each image pair, given a real image (left) and a text prompt, our method (right) facilitates *zero-shot text-guided editing* without requiring fine-tuning of Stable Diffusion. Our results exhibit complex, non-rigid, consistent, and faithful editing while preserving the structure and scene layout in the original image. Our proposed method addresses various image editing tasks, including object replacement (left column), object removal and background alteration (middle column), the addition of new consistent items, and changes in object pose/view (right column).

## Abstract

Editing natural images using textual descriptions in text-to-image diffusion models remains a significant challenge, particularly in achieving consistent generation and handling complex, non-rigid objects. Existing methods often struggle to preserve textures and identity, require extensive fine-tuning, and exhibit limitations in editing specific spatial regions or objects while retaining background details. This paper proposes *Context-Preserving Adaptive Manipulation (CPAM)* – a novel zero-shot method for complicated, non-rigid real image editing. Specifically, we propose a preservation adaptation module that adjusts self-attention mechanisms to preserve and independently control the object and background effectively. This ensures that the objects' shapes, textures, and identities are maintained while keeping the background undistorted during the editing process using the mask guidance technique. Additionally, we develop a localized extraction module to mitigate the interference with the non-desired modified regions during conditioning in cross-attention mechanisms. We also introduce various mask-guidance strategies to facilitate diverse image manipulation tasks in a simple manner. Extensive experiments on our newly constructed Image Manipulation BenchmArk (IMBA), a robust benchmark dataset specifically designed for real image editing, demonstrate that our proposed method is the preferred choice among human raters, outperforming existing state-of-the-art editing techniques.

## 1 Introduction

Recent advancements in text-to-image (T2I) generation  (Ramesh et al., 2021; Dhariwal & Nichol, 2021; Nichol et al., 2022; Yu et al., 2022; Ramesh et al., 2022; Saharia et al., 2022) have marked

significant milestones, especially with large-scale diffusion models Rombach et al. (2022) that excel in creating diverse and high-quality images from text prompts. These models have opened new avenues for text-conditioned image editing (Hertz et al., 2023; Tumanyan et al., 2023; Parmar et al., 2023).

Real image editing typically aims to produce multiple images of different complex, non-rigid objects or characters that resemble the original targeted object while also ensuring a perfect reconstruction of the background (Vo et al., 2024; Wallace et al., 2023; Pan et al., 2023; Parmar et al., 2023). However, this presents notable challenges. Text-guided editing of a real image using state-of-the-art diffusion models (Kim et al., 2022) requires inverting the given image, which involves finding an initial latent noise that accurately reconstructs the input image while preserving the model's editing capabilities. Editing an image from that latent noise often results in losing original textures and identity, leading to a different image. Additionally, existing methods are limited in editing specific objects within images, as they often focus on the most salient objects. This limitation arises from training diffusion models (Rombach et al., 2022) on image-captioning datasets (Schuhmann et al., 2021; 2022), which may lack detailed descriptions of text prompts for real-world images. Thus, pre-trained Stable Diffusion (SD) is unable to focus on specific regions and instead operates on the overall image. Furthermore, real-world images often contain multiple objects and complex interactions, making it challenging to specify particular objects for editing. Additionally, fully fine-tuning large models like SD (Rombach et al., 2022) is less feasible in research areas with limited computational resources.

To address the lack of facilities for training models on large-scale datasets, tuning-free methods have been developed to utilize pre-trained T2I SD (Rombach et al., 2022), referred to as *zero-shot image editing*. These methods leverage a pre-trained T2I model with frozen weights to eliminate the need for adjusting the model's weights (Avrahami et al., 2022; Meng et al., 2022; Brack et al., 2024). Most methods (Hertz et al., 2023; Cao et al., 2023; Liu et al., 2024; Parmar et al., 2023; Tumanyan et al., 2023) rely on attention mechanisms in SD models to preserve the original information of images, such as background and object identities. Specifically, some methods (Tumanyan et al., 2023; Liu et al., 2024) swap or inject appropriate self-attention maps, while others, like Hertz et al. (2023), replace cross-attention maps to retain the content and structure of the original image during the synthesis process. However, these methods (Tumanyan et al., 2023; Liu et al., 2024) perform well when the edited object has a certain similarity to the original object in terms of shape, texture, and other attributes. Similarly, the approach in Hertz et al. (2023) that replaces cross-attention maps requires the initial prompt and edited prompt to share similar words while incorporating different words. For instance, if the original sentence is 'the photo of a red dog', the edited sentence might be 'the photo of a yellow cat', where 'red dog' and 'yellow cat' are the differing elements. In contrast, Cao et al. (2023) adjusts self-attention to retain the current query features while replacing the key and value features. This approach ensures that the query features remain unchanged and are appropriately derived from the original semantic content guided by masks, rather than relying on rigidly swapped attention maps. As a result, Cao et al. (2023) preserves the appearance of the original image in a non-rigid manner during synthesis. However, Cao et al. (2023) controls the background and foreground simultaneously to obtain the semantic content of the corresponding original background and foreground at each appropriate step and layer. Thus, this approach lacks flexibility in controlling different image editing tasks; for example, we need the background to remain unchanged when the edited object resembles the original object. Additionally, a significant weakness of many methods is that, when editing images, they make changes to the overall image and cannot specifically edit individual objects within the image due to the condition of all image pixels and the text prompt in the cross-attention module (as shown in the middle images of Figure 4). Some methods (Hertz et al., 2023; Avrahami et al., 2022; 2023; Couairon et al., 2023) address the challenge of local editing by blending the original latent noise with the edited noise, without considering the interaction between foreground and background, resulting in rigid editing. Subsequently, these methods lead to a substantial gap in addressing real image editing tasks.

Based on the existing extensive exploration of leveraging the attention modules in SD to control the editing process and achieve desired outcomes, we analyze and clarify the semantic interaction of the components in attention and how to leverage them for real image editing (as detailed in Section. 3.1). We then propose a novel zero-shot real image editing method, namely **C**ontext-**P**reserving **A**daptive **M**anipulation (CPAM), that leverages both self-attention and cross-attention. Our method excels in manipulating non-rigid objects, allowing for modifications to various aspects such as pose,

view, or even specific objects or parts within the image. Importantly, our CPAM retains the background and avoids modifications to unwanted objects or regions, thereby addressing issues faced by existing methods, enabling object removal or background replacement (as illustrated in Figure. 1). Notably, these modifications occur without any model configuration or system architecture adjustments, eliminating the need for optimization or fine-tuning phases. Specifically, we introduce a preservation adaptation process that adjusts self-attention to independently control the object and background, effectively preserving the original objects' shapes, textures, and identities using masking techniques. Simultaneously, it ensures that the background remains undistorted or unwarped throughout the denoising process. Additionally, we propose the localized extraction module to avoid attention between the non-desired modified regions with the target prompt in the cross-attention. Therefore, our method enables localized editing, allowing for editing not only salient objects but also specific objects within the image. We propose different mask-guidance strategies to enable innovative image editing tasks by simply adjusting masks and enhancing regional manipulation by controlling object shapes. The source mask, representing the original object, and the target mask, controlling the edited outcome, are computed differently, allowing for flexible image editing tasks.

In addition, we introduce a new **I**mage **M**anipulation **B**enchm**A**rk (IMBA), built upon TEd-Bench (Kawar et al., 2023). We conduct a comprehensive user study on IMBA to assess the performance of our method against state-of-the-art text-guided image editing techniques utilizing SD. The extensive experimental results unequivocally highlight the superiority of our proposed method, significantly outperforming state-of-the-art methods. Our contributions can be summarized as follows:

- We propose a novel tuning-free method dubbed **C**ontext-**P**reserving **A**daptive **M**anipulation that leverages both self-attention and cross-attention for zero-shot real image editing.

- We propose the preservation adaptation process to control and preserve various aspects of objects such as pose, view, texture, identities, structures, color, and non-rigid variances while retaining the background.

- We propose the localized extraction module to prevent any unwanted effects of the target prompt on the non-desired modified spatial region in cross-attention.

- We present mask-guidance strategies to facilitate various image manipulation tasks simply, while also tracking object shapes during the synthesis process.

- We construct a new Image Manipulation BenchmArk (IMBA) dataset to contain more desired information for real image editing.

## 2 RELATED WORK

### 2.1 IMAGE MANIPULATION METHODS

Several approaches required optimization or fine-tuning phases, which self-learned input images (Mokady et al., 2023; Kawar et al., 2023). Ruiz et al. (2023) synthesized novel views of a given subject using 3–5 images of that subject and a target prompt. Gal et al. (2023) optimized a new word embedding token for each concept. Kawar et al. (2023) generated novel poses and views by optimizing the target text embedding, fine-tuning model parameters, and interpolating between the approximate and target text embeddings. However, it struggled to maintain background consistency and realism, requiring careful optimization of embeddings for each prompt-image pair. Null-Text Inversion (NULL)(Mokady et al., 2023) proposed optimal image-specific null-text embeddings for accurate reconstruction, combined with P2P(Hertz et al., 2023) techniques for real image editing. Brooks et al. (2023) performed full fine-tuning of the diffusion model by generating image-text-image triplets based on instructional input. However, the optimization and fine-tuning process is time-consuming and resource-intensive. Our method, instead, focuses on tuning-free techniques that eliminate the need for such processes.

### 2.2 ZERO-SHOT METHODS

Zero-shot approaches focused on editing images directly during the denoising phase, eliminating the need for any fine-tuning or additional training. SDEdit (Meng et al., 2022) introduced intermediate

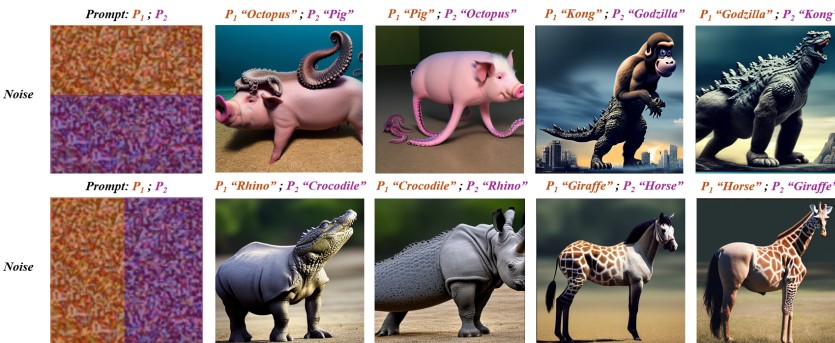

Figure 2: We perform multi-text guided synthesis, where each text prompt conditions a distinct part of the latent noise, effectively leading to results that exceed expectations and demonstrating the ability to condition each part with its respective prompt. Please zoom in for a clearer view.

noise to an image, followed by denoising through a diffusion process conditioned on the desired edit. However, it exhibited a tradeoff between preserving the original image attributes and fully achieving the target text's intended changes. Blended Diffusion (Avrahami et al., 2022) facilitated local editing using gradient guidance based on the CLIP loss of the desired modified region and the target text prompt, without accounting for the interaction between foreground and background. However, blending this with the original image noise at each step led to rigid editing and inconsistency. Chefer et al. (2023) generate images that fully convey the semantics of the given text prompt by progressively guiding the noised latent at each timestep, using the attention maps of the subject tokens from the prompt. Brack et al. (2024) proposed approaches for quickly and accurately inverting images and determining the appropriate direction for editing. Parmar et al. (2023) requires a large bank of diverse sentences from both source and target texts to form an edit direction. Huberman-Spiegelglas et al. (2024) introduced an inversion method for DDPM, showing that the inversion maps encoded the image structure more effectively than the noise maps used in regular sampling, making them better suited for image editing.

## 3 PROPOSED METHOD

### 3.1 PRELIMINARY ANALYSIS OF ATTENTION MECHANISM IN STABLE DIFFUSION

Within the Stable Diffusion (SD) Rombach et al. (2022), the attention mechanism Vaswani et al. (2017) of the denoising U-Net, which includes both self-attention and cross-attention, is mathematically expressed as $\text{Attention}(Q, K, V) = \text{Softmax}\left(\frac{QK^T}{\sqrt{d}}\right)V$, where $Q$ represents the query features projected from spatial features, while $K$ and $V$ are the key and value features projected from spatial features (in self-attention layers) or textual embeddings (in cross-attention layers) using the respective projection matrices.

**Insights from cross-attention** Cross-attention involves interactions between pixels and prompts (i.e., key and value features from textual embeddings). First, we observed that attending each prompt to different parts of latent noise allows each section to be conditioned by its respective prompt (as depicted in Figure. 2). Second, null text does not affect the output, a phenomenon evident during training. Most diffusion models (DMs) utilize a classifier-free guidance (Ho & Salimans, 2021), randomly replacing text conditioning with null text at a fixed probability during training. As a result, when latent noise parts attend to null text, the corresponding pixels are perfectly reconstructed. Our method leverages this by directing attention to the pixels of specific objects using the text prompt, while background pixels attend to null text.

**Insights from self-attention** Previous works (Cao et al., 2023; Liu et al., 2024; Tumanyan et al., 2023) show that self-attention features can be injected into U-Net layers for image translation, preserving semantic layout. Our key insight is that self-attention lets pixels connect with themselves, creating smooth transitions and consistent interactions. For example, in Figure. 2, we apply two

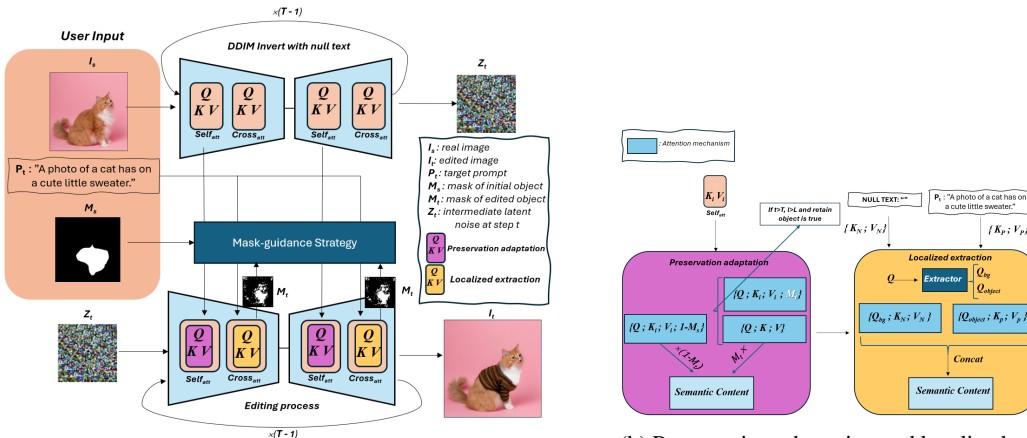

(a) CPAM pipeline.

(b) Preservation adaptation and localized extraction modules.

Figure 3: The architecture of our proposed Context-Preserving Adaptive Manipulation (CPAM) for zero-shot real image editing consists of several key steps. First, we invert the image into latent noise using the deterministic inversion technique of DDIM (Song et al., 2021) with null-text guided. During the editing process, we utilize a preservation adaptation module to maintain the original attributes while mitigating effects in the background through a localized extraction module. Please zoom in for a clearer view.

---

**Algorithm 1** Zero-Shot Real Image Editing

---

**Inputs:**
A target prompt $P_t$, A source mask $M_s$
The intermediate latent noises $z_i$, the target initial latent noise map $z_T$
**Output:** edited latent map $z_0$

1: **for** $t = T, T-1, \ldots, 1$ **do**
2:     $M_t \leftarrow$ Mask-guidance strategy(cross-attention maps, $M_s$)
3:     $\{\_, K_i, V_i\} \leftarrow \epsilon_\theta(z_i, t)$
4:     $\{Q, K, V\} \leftarrow \epsilon_\theta(z_t, t)$
5:     inputs $\leftarrow (Q, K, V, K_i, V_i, M_s, M_t)$
6:     self-attention $\overset{\text{adapt}}{\longleftarrow}$ Preserving-adaptation(inputs)
7:     cross-attention $\overset{\text{inject}}{\longleftarrow}$ Localized-extraction($Q, P_t, P_{nulltext}, M_t$)
8:     $\epsilon \leftarrow \epsilon_\theta(z_t, P_t, t, \text{self-attention}, \text{cross-attention})$
9:     $z_{t-1} \leftarrow \text{Sample}(z_t, \epsilon)$
10: **end for**
11: **return** $z_0$

---

prompts (e.g., 'crocodile' and 'rhino') to two parts of latent noise, resulting in a cohesive and non-rigid outcome. Self-attention also helps each pixel determine which others to attend to, even when excluding a specific region, as shown in Figure 1, where all image pixels focus on the background pixels, excluding the teddy bear pixels, effectively removing it without disrupting the connections of the semantic in the image. The process of object removal is further explained in Section A.1. By controlling self-attention, we minimize the impact on irrelevant areas while preserving image coherence.

## 3.2 OVERVIEW

Based on the insights in Section. 3.1, we propose Context-Preserving Adaptive Manipulation (CPAM) to edit an image $I_s$ using a source object mask $M_s$ and a target text prompt $P_t$ to generate a new image $I_t$ that aligns with $P_t$. Notably, $I_t$ may spatially differ from $I_s$, modifying objects or background while keeping other regions unchanged. To achieve this, we introduce a preservation adaptation module that adjusts self-attention to align the semantic content from intermediate latent noise to the current edited noise, ensuring the retention of the original object and background during

the editing process. To prevent unwanted changes from the target prompt in non-desired modified regions, we propose a localized extraction module that enables targeted editing while preserving the remaining details. Additionally, we propose mask-guidance strategies for diverse image manipulation tasks. The overall CPAM architecture is illustrated in Figure. 3a, and the zero-shot editing algorithm is outlined in Algorithm 1.

## 3.3 PRESERVATION ADAPTATION

In this subsection, we describe the self-attention adaptation process, which preserves the original image's appearance by independently adapting the semantic content from intermediate latent noise to the edited image.

**Background preservation adaptation**   To adapt semantic content from intermediate latent noise during the denoising step $t$, we retain the query features $Q$ and extract the original semantic content from the key and value features $K$ and $V$ at that self-attention layer. We then apply attention guided by the mask $M_s$. The semantic content of the background $SC_{bg}$ can be formulated as:

$$SC_{bg} = \text{Att}(Q, K_i, V_i; 1 - M_s), \tag{1}$$

where $K_i$ and $V_i$ correspond to the key and value features of the intermediate latent noise, respectively, and Att is the attention mechanism.

**Object preservation adaptation**   Preserving the object's semantic content is more difficult than maintaining the background because it requires adapting the original object's features to fit a new shape, pose, or view. We carefully control this adaptation after $S$ step and layer $L$, while generating new shapes, poses, or views. Thus, the semantic content of the object (foreground) $SC_{fg}$ can be formulated as follows:

$$SC_{fg} = \begin{cases} \text{Att}(Q, K_i, V_i; M_t), & \text{If } t > T \text{ and } l > L \text{ and the object is retained,} \\ \text{Att}(Q, K, V), & \text{otherwise,} \end{cases} \tag{2}$$

where $t, l, T = 3, L = 8$ denote step and layer, respectively. $V_i$ are the value feature of intermediate latent noise at step $i$. $K$ and $V$ are the key and value features of current noise, respectively. $M_t$ is the target mask of the edited object, and Att is the attention mechanism.

**Location adaptation**   The aim of this module is to provide precise control over the foreground and background during image editing, allowing for independent adjustments. By separately deriving the semantic content of the background from Equation. 1 and the foreground from Equation. 2, and aligning them with the target mask $M_t$, we achieve flexible, region-specific edits. This process ensures that modifications occur only in designated areas, keeping the rest of the image intact. Consequently, the combined semantic content $SC$, guided by the target mask, is formulated as follows:

$$SC = M_t \odot SC_{\text{foreground}} + (1 - M_t) \odot SC_{\text{background}}, \tag{3}$$

where $\odot$ denotes the element-wise multiplication.

However, applying self-attention independently to the foreground and background leads to a lack of interaction, resulting in rigid editing. To maintain overall image coherence, we randomly apply normal self-attention in 10% of the layers. This approach minimizes unintended distortions and yields more natural results during synthesis.

## 3.4 LOCALIZED EXTRACTION

Despite significant efforts to adapt the content of the original image in our preservation adaptation module, the non-desired modified spatial region may appear distorted (as shown in the third image in Figure. 4). This distortion arises because all pixels of the image attend to tokens of the text prompt, affecting both the foreground (i.e., object) and background, including non-desired modified regions. To address this issue, we introduce a localized extraction mechanism, allowing for editing only a specific object without distorting the rest. This mechanism applies attention to the extracted object's spatial pixels from the feature query to the target prompt, while the remaining pixels attend to the null text prompt:

$$\text{LE}(Q, K, V) = \text{Att}(\text{Extract}(Q, M_t), K_t, V_t) \oplus \text{Att}(\text{Extract}(Q, 1 - M_t), \text{Null}(K), \text{Null}(V)), \tag{4}$$

Figure 4: The real image (left), along with the edit prompt "*Messi and a rugby*," and a mask suggests a desire to transform the soccer ball into a rugby ball. In two middle images, MasaCtrl (Cao et al., 2023) produces another instance of Messi without mask guidance and retains the background with mask guidance, but the problem remains unsolved (marked by blue annotation). On the other hand, the localized extraction in our CPAM can successfully preserve the background while transforming the soccer ball into the rugby ball (the right image). Please zoom in for a clearer view.

where $\oplus$ represents concatenation, $\mathrm{Extract}(Q, M)$ extracts the object's spatial pixels from the feature query $Q$ where $M = 1$, and Att is the attention mechanism.

### 3.5 Mask-guidance strategy

Achieving both rigid and non-rigid semantic changes within a unified framework for diverse image manipulation tasks is a notable challenge. Our method is designed to simplify this by requiring only adjustments to mask settings. We present various mask guidance strategies tailored to different editing needs. The source mask $M_s$ refers to the mask of the object or region the user wishes to edit, which can be provided through various methods such as manual drawing, extraction from clicks, or text prompts using SAM (Kirillov et al., 2023). Meanwhile, the target mask $M_t$ can be obtained as follows:

- Replacing an object or changing the object pose, view: $M_t$ is achieved by aggregating cross-attention maps across all steps and layers or extending the convex hull of $M_s$.

- Altering background: When $M_s$ is the mask of the background, we assign $M_t = M_s$.

- Removing object: We assign $M_t = 0$, meaning in our preservation adaptation module only adapting the semantic content of the original background (refer to Equation 3).

- Modifying a specific spatial region (e.g., adding items): We simply assign $M_t = M_s$ or slightly expand $M_s$.

**Mask refinement**   When manipulating an object, its shape may change during diffusion steps. To address this, we refine the mask according to the target prompt during the denoising process. The target mask $M_t$ is automatically obtained by aggregating cross-attention maps. Initially, for the first $T_m$ steps, we use the source mask $M_s$, then transition to $M_t$, which can be cloned from $M_s$ or derived from the generation process. Additionally, we avoid closely segmenting both the target and original objects to prevent overly rigid editing and the leakage of underlying shape information.

## 4 Experiments

### 4.1 Implementation details

All experiments were conducted on a machine with a single A100 GPU. Our proposed CPAM was employed using the publicly available SD-1.5 model. We initially encode the image into latent code by variational autoencoder (Kingma & Welling, 2014) and invert to noise using the deterministic inversion technique of DDIM (Song et al., 2021) with null-text guided. In the sampling process, we employed DDIM sampling by with 50 denoising iterations, and the classifier-free guidance was set at 7.5.

Table 1: Comparison of state-of-the-art methods using FID assesses image quality, CLIPScore measures text-image alignment, LPIPS (background) evaluates background preservation, and Inception Score reflects diversity and realism. FID, LPIPS: lower is better ↓; CLIPS, IS: higher is better ↑.

| Method | FID ↓ | CLIPScore ↑ | LPIPS (background) ↓ | IS ↑ |
|---|---|---|---|---|
| SDEdit (Meng et al., 2022) | 180.37 | 28.19 | 0.338 | 33.33 |
| MasaCtrl (Cao et al., 2023) | 101.05 | 28.82 | 0.223 | 49.32 |
| PnP (Tumanyan et al., 2023) | 89.00 | 29.03 | 0.162 | 89.50 |
| FPE (Liu et al., 2024) | **75.90** | 29.02 | 0.152 | **92.97** |
| DiffEdit (Couairon et al., 2023) | 90.77 | 28.58 | 0.148 | 48.77 |
| Pix2Pix-Zero (Parmar et al., 2023) | 122.53 | 27.01 | 0.186 | 22.82 |
| LEDITS++ (Brack et al., 2024) | 92.93 | 28.74 | **0.141** | 41.03 |
| Imagic (Kawar et al., 2023) | 123.41 | **30.34** | 0.420 | 47.14 |
| **CPAM (Ours)** | 93.34 | **29.26** | **0.149** | 43.11 |

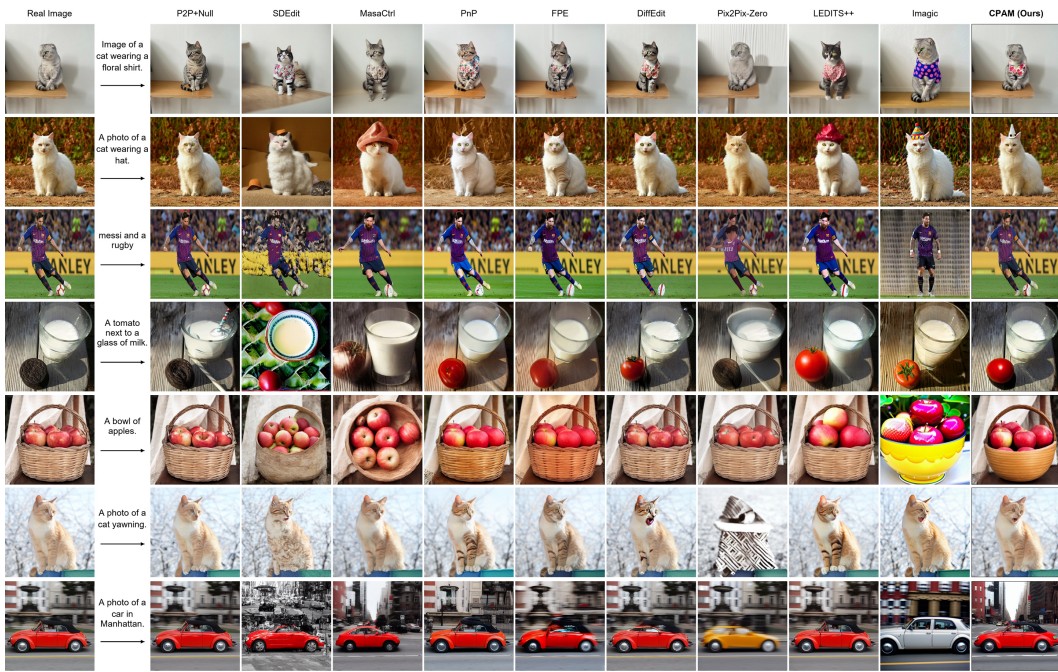

Figure 5: Qualitative comparison of our proposed CPAM method with state-of-the-art approaches. Our CPAM outperforms existing methods across multiple real image editing tasks. Please zoom in for a clearer view.

## 4.2 IMAGE MANIPULATION BENCHMARK (IMBA)

Textual Editing Benchmark (TEdBench) Kawar et al. (2023) was the pioneered standard benchmark for assessing non-rigid text-based real image editing. The dataset comprises 100 pairs of input images and target texts, describing complex non-rigid edits. However, the dataset lacks detailed user-editing preferences, such as object retaining, background altering, etc, and lacks evaluation on specific object editing. Thus, we introduced Image Manipulation BenchmArk (IMBA) to address these limitations, built upon the TEdBench. IMBA dataset not only incorporates detailed user-editing preferences but also includes additional inputs, such as alteration masks and object prompts, to enhance control during the editing process. Moreover, IMBA adds samples containing multiple objects, facilitating the evaluation of specific object editing capabilities. In total, IMBA includes 104 samples, with 43 samples requiring object retention, 97 samples involving object modification, and 7 samples involving background alteration.

Table 2: User study results measuring participants' opinion (1: very bad, 6: very good) in rating image editing methods. Our CPAM significantly outperforms existing methods.

| Method | Object Retention | Background Retention | Realistic | Satisfaction |
|---|---|---|---|---|
| SDEdit (Meng et al., 2022) | 3.63 | 3.19 | 3.38 | 2.42 |
| MasaCtrl (Cao et al., 2023) | 4.01 | 4.17 | 4.32 | 3.11 |
| PnP (Tumanyan et al., 2023) | 4.61 | 4.49 | 4.20 | 2.63 |
| FPE (Liu et al., 2024) | 4.50 | 4.44 | 4.33 | 2.53 |
| DiffEdit (Couairon et al., 2023) | 4.58 | 4.57 | 4.40 | 3.13 |
| Pix2Pix-Zero (Parmar et al., 2023) | 2.11 | 4.23 | 1.84 | 1.93 |
| LEDIT++ (Brack et al., 2024) | 4.38 | 4.95 | 4.57 | 3.26 |
| Imagic (Kawar et al., 2023) | 3.74 | 3.48 | 4.30 | **4.82** |
| **CPAM (Ours)** | **4.72** | **5.09** | **4.69** | 3.30 |

## 4.3 QUALITATIVE AND QUANTITATIVE EVALUATION

In this section, we present both qualitative and quantitative assessments of our proposed CPAM method in comparison to state-of-the-art image editing approaches.

Figure 5 provides a qualitative comparison of our CPAM method against leading techniques in image editing based on Stable Diffusion (SD), such as P2P+NULL (Hertz et al., 2023; Mokady et al., 2023), SDEdit (Meng et al., 2022), MasaCtrl (Cao et al., 2023), PnP (Tumanyan et al., 2023), FPE (Liu et al., 2024), DiffEdit (Couairon et al., 2023), Pix2Pix-Zero (Parmar et al., 2023), LEDIT++ (Brack et al., 2024), and the fine-tuning method Imagic (Kawar et al., 2023). Our results indicate that CPAM consistently outperforms these existing methods across various real image editing tasks. This performance is particularly evident in its ability to modify diverse aspects of images, including pose, view, background changes, and specific object alterations, all while effectively preserving the original background and avoiding unintended modifications.

In Table 1, we compare the quantitative metrics of our CPAM method against other state-of-the-art approaches. We excluded the evaluation of P2P (Hertz et al., 2023) combined with NULL (Mokady et al., 2023) due to its reliance on an initial prompt that often leads to unchanged outputs. The data reveals a clear trend: while SDEdit and other methods struggle to maintain structural integrity and background details, our CPAM method achieves high CLIP accuracy alongside low structure distortion and background LPIPS scores. This combination demonstrates our capability to execute edits effectively while retaining the essential features of the original input images.

Methods like SDEdit (Meng et al., 2022) often yield unrealistic results due to their dependence on noise strength parameters, which can disrupt semantic consistency. MasaCtrl (Cao et al., 2023) lacks precise control over background and foreground elements during denoising, leading to unwanted alterations. PnP (Tumanyan et al., 2023) preserves the background but often fails to meet the target prompt, while FPE (Liu et al., 2024) generates minimal visible changes due to its high reliance on self-attention maps. Pix2Pix-Zero (Parmar et al., 2023) struggles in real image editing tasks due to its dependence on closely matched prompts. Additionally, DiffEdit (Couairon et al., 2023) and LEDIT++ (Brack et al., 2024) often capture the entire object when generating masks based on noise estimation, resulting in unwanted modifications. Although Imagic (Kawar et al., 2023) excels in user satisfaction, it frequently struggles with background retention and can produce misalignments or unwanted alterations, and requires more time consuming for fine-tuning and optimizing for each image prompt pair.

In contrast, our CPAM method demonstrates a more robust performance in preserving both object integrity and background details, effectively executing complex edits without sacrificing quality. This combination of qualitative and quantitative evaluations underscores the effectiveness of our approach in the context of modern image editing techniques.

## 4.4 USER STUDY

To further assess the effectiveness of our proposed CPAM method, we conducted a user study comparing it against several leading prompt-based editing methods utilizing diffusion models. The methods evaluated include SDEdit (Meng et al., 2022), MasaCtrl (Cao et al., 2023), PnP (Tumanyan et al.,

2023), FPE (Liu et al., 2024), DiffEdit (Couairon et al., 2023), Pix2Pix-Zero (Parmar et al., 2023), LEDIT++ (Brack et al., 2024), and the fine-tuning method Imagic (Kawar et al., 2023).

To ensure a comprehensive evaluation, we defined four key metrics: object retention, background retention, realism, and overall satisfaction. These metrics are designed to assess the methods' effectiveness in executing realistic edits while preserving important features of the original images:

- Object Retention: This metric evaluates how well the method preserves the identity and details of the main object in the image during editing.

- Background Retention: This assesses the method's ability to maintain the integrity and appearance of the background while altering the main object.

- Realism: This metric analyzes the realism of the edits, particularly in the context of non-rigid transformations.

- Satisfaction: This measures the degree to which the edited image aligns with the textual description provided as the editing prompt.

We invited 20 participants from diverse professional backgrounds to provide a variety of perspectives in the evaluation process. Each participant rated the performance of the methods on a scale from 1 (very bad) to 6 (very good) across the four metrics. Participants evaluated 50 randomly shuffled images for each method, resulting in a total of 36,000 responses.

Table 2 presents the Mean Opinion Score (MOS) derived from the participants' ratings. The results demonstrate that our CPAM method significantly outperforms the other methods across most metrics. Notably, CPAM received the highest ratings for object retention, background retention, and realism, indicating its superior ability to maintain key elements of the images while executing edits effectively. While Imagic (Kawar et al., 2023) excelled in user satisfaction but it faced challenges in background retention, occasionally produced unrealistic outputs, and required significantly more time for fine-tuning and optimization for each image-prompt pair.

Overall, the user study reinforced the findings from our qualitative and quantitative evaluations, highlighting the effectiveness of CPAM in real image editing tasks.

### 4.5 LIMITATIONS AND DISSCUSION

Similar to other zero-shot methods, our method is constrained by the capabilities of the pre-trained model. Sometimes, the generated images may not align perfectly with the provided prompts. Our mask refinement module aims to obtain masks from cross-attention maps by applying a simple standardization technique. However, this approach may result in imprecise object shapes or may overly focus on prominent objects, leading to suboptimal outcomes. While we can address this issue by adjusting or slightly expanding the masks, there are situations where these solutions may not be sufficient. When editing a specific spatial region of an image, precise prompts and initial masks are necessary, and the model must generate content in that region. Unfortunately, we may encounter difficulties editing small or non-salient objects or when content cannot be generated in that region. This challenge arises because the SD model is primarily trained on image-captioning datasets, where the text prompts typically focus on salient objects. Fortunately, our method allows for an increased guidance scale, effectively addressing this challenge and enhancing the model's ability to generate content in less prominent areas.

### 5 CONCLUSION

Our CPAM facilitates various zero-shot real image editing tasks by leveraging both self-attention and cross-attention mechanisms within SD models. Overcoming existing limitations, CPAM employs a preservation adaptation process to meticulously control and retain various object attributes while preserving the background. Additionally, our method features a localized extraction module to prevent undesired effects of target prompts on non-desired spatial regions, enabling precise object editing within images. We also introduce IMBA dataset, providing rich information for comprehensive image manipulation evaluations. Empirical results demonstrate that our CPAM consistently outperforms existing leading editing techniques in achieving complicated and non-rigid edits.

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

# A APPENDIX

## A.1 ABLATION STUDY

Most diffusion models (DMs) rely on classifier-free guidance (Ho & Salimans, 2021), where a low guidance scale can produce overly abstract or unrelated images, while a high guidance scale can result in images that look rigid or unnatural, as the model over-commits to the prompt, potentially sacrificing creativity and naturalness. However, our method leverages a higher guidance scale to edit images without causing distortion. It achieves this using simple prompts, avoiding the need for complex and precise text descriptions, making the editing process more intuitive and user-friendly (as shown in Figure 6).

We further explain that our method enables object removal. To achieve this during the editing process, we control self-attention to ensure that all spatial pixels attend only to the background while disregarding the object content. This effectively removes the object without breaking the semantic structure of the image, as demonstrated in Figure 7.

## A.2 COMPARATIVE PIPELINE ANALYSIS: CPAM VS MASACTRL

In our comparative analysis, we examine the intricacies of CPAM and MasaCtrl, as illustrated in Figure. 8. Unlike MasaCtrl, which lacks independently control over the semantic content of the background and object across different steps and layers and CPAM employs localized extraction in contrast to MasaCtrl's employment of normal cross-attention.

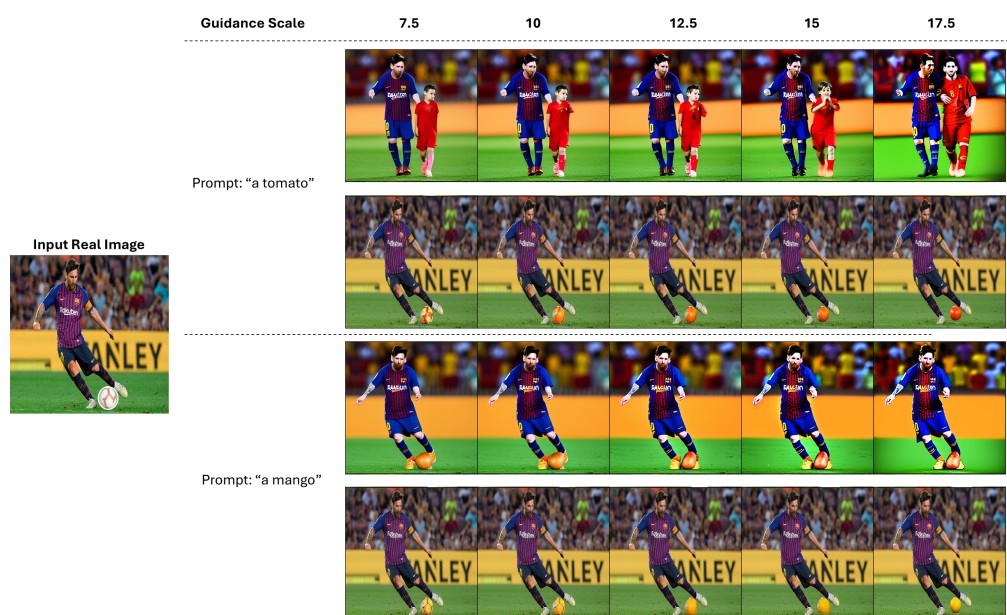

Figure 6: We demonstrate the effect of image consistency and control with and without using our method. Our approach allows for a high guidance scale, achieving the desired results without distorting the image, requiring only a simple prompt from the user.

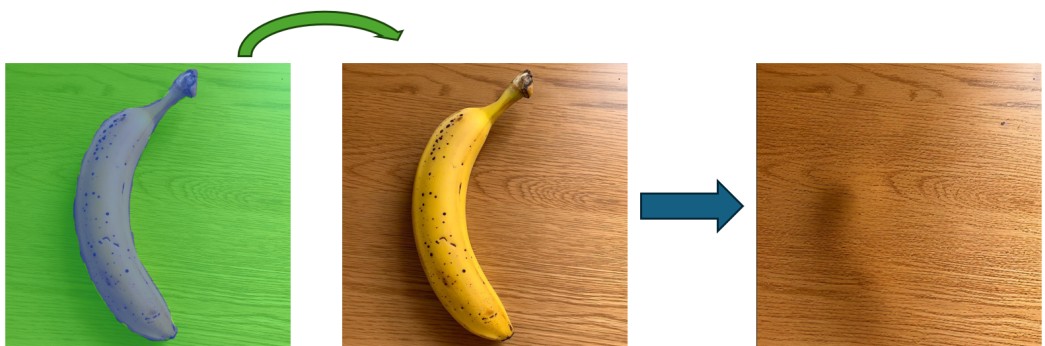

Figure 7: Our method controls self-attention to focus only on the background of the image, marked by the green region, effectively removing the object.

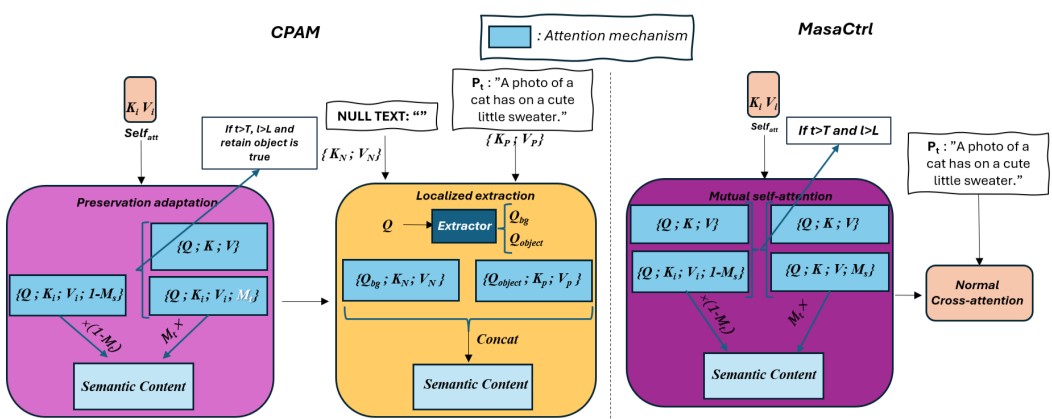

Figure 8: The Comparative Details of CPAM and MasaCtrl.

## A.3 IMAGE MANIPULATION BENCHMARK (IMBA) DETAILS

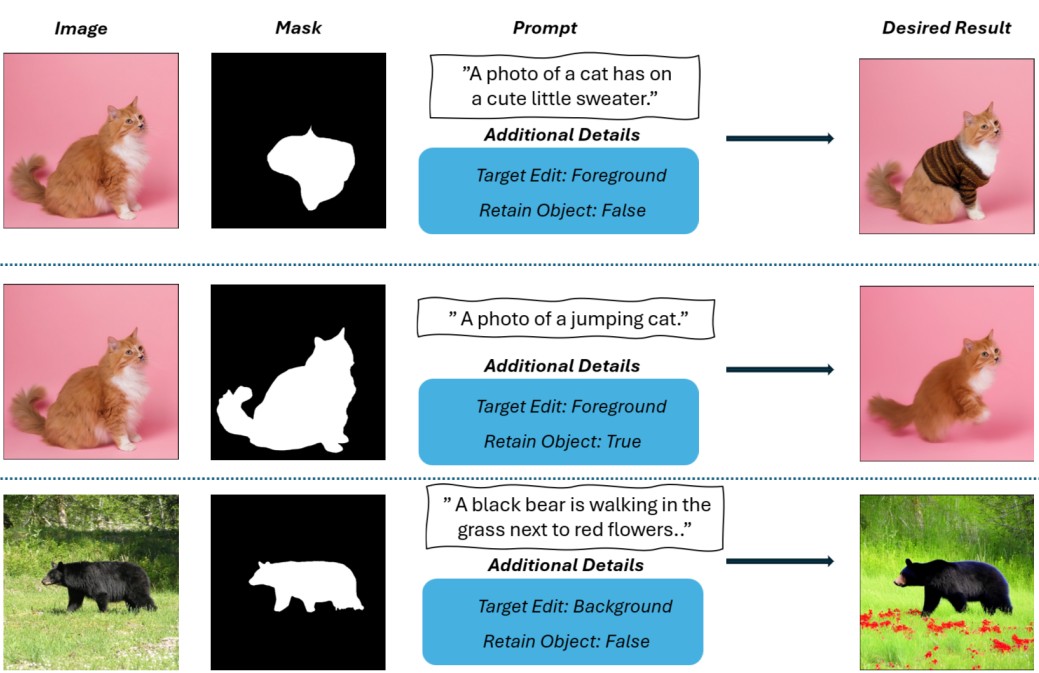

Figure 9: The visualization of Image Manipulation BenchmArk (IMBA).

We introduced Image Manipulation BenchmArk (IMBA) built upon the TEdBench. IMBA dataset not only incorporates detailed user-editing preferences but also includes additional inputs, such as alteration masks and object prompts, to enhance control during the editing process (as illustrated in Figure. 9).

## A.4 FURTHER EXAMPLES

We provide additional visualizations for a more qualitative evaluation (Figure 10) and various real image manipulation tasks, including removing objects (Figure. 11),

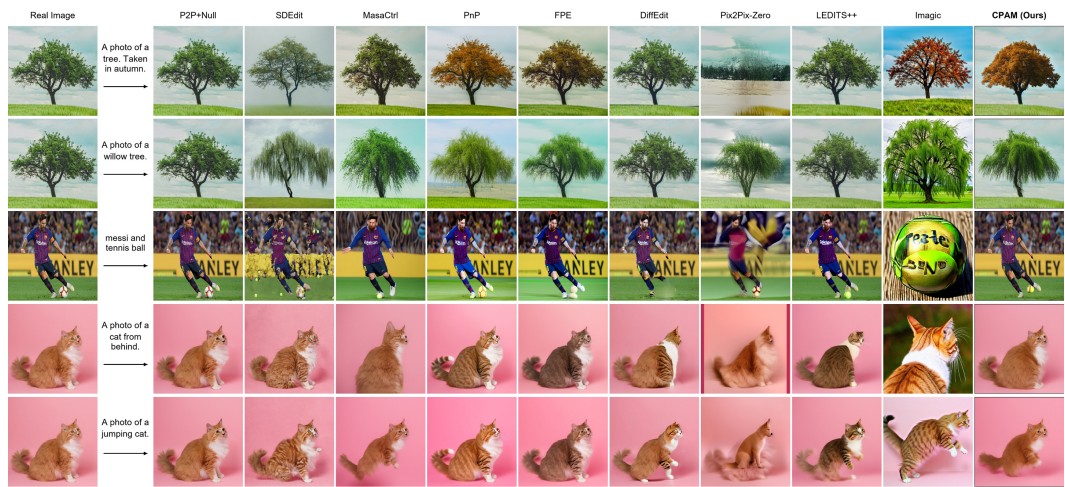

Figure 10: More qualitative results comparing our proposed CPAM method with state-of-the-art techniques.

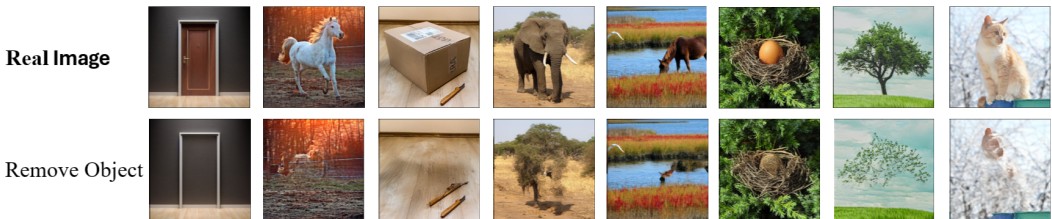

Figure 11: CPAM effectively removes the object.

## A.5 EXPERIMENTAL DETAILS

All experiments were conducted using Stable Diffusion 1.5 and guidance scale 7.5, 50 inference steps. For all methods, we utilized publicly available official code, with the exception of Imagic (Kawar et al., 2023). For Imagic, we evaluated publicly available results and leveraged community-developed code from the Diffusers library on Hugging Face.

We performed a grid search across the hyperparameter ranges specified for each method while keeping other parameters at their default settings. Initially, a wider range of values was explored to define reasonable boundaries, after which edge values that resulted in performance declines were discarded.

**P2P+NULL (Hertz et al., 2023; Mokady et al., 2023)** Get initial prompt by Clip (Radford et al., 2021), with the rate of replacing self-attention steps set between 0.4 and 0.7.

**SDEdit (Meng et al., 2022)** Diffusion steps between 25 (with strength 0.5 at 50 steps) and 40 steps (with strength 0.8 at the default 50 steps).

**MasaCtrl (Cao et al., 2023)** Step query set to 4, layer query between 10 and 14, with three mask options: no mask guidance, explicit mask, and auto-aggregated mask.

**PnP (Tumanyan et al., 2023)** The rate of replacing self-attention steps and feature injection between 0.5 and 0.8, 50 inference steps.

**FPE (Liu et al., 2024)** The rate of replacing self-attention steps is set between 0.5 and 0.8.

**DiffEdit (Couairon et al., 2023)**  Get initial prompt by Clip (Radford et al., 2021).

**Pix2Pix-Zero (Parmar et al., 2023)**  Generate 5 source prompts and 5 target prompts by flan-t5-xl model (Chung et al., 2024).

**LEDIT++ (Brack et al., 2024)**  50 inversion steps, skip between 0.1 and 0.3.

**Imagic (Kawar et al., 2023)**  500 text embedding optimization steps, 1000 model finetuning steps, $\alpha$ between 0.1 and 2.0.

A.6  USER STUDY DETAILS

**Participants**  We invited 20 participants (17 males and three females, age $\in [16, 22]$) from our research community, including students with knowledge about AI and those from outside the industry, to participate in our study. All participants were new to AI generative tasks, although some had previously participated in various user studies related to AI. With diverse professional backgrounds, they brought different perspectives to the evaluation process, ensuring an objective assessment (see an overview of the information of participants in  12).

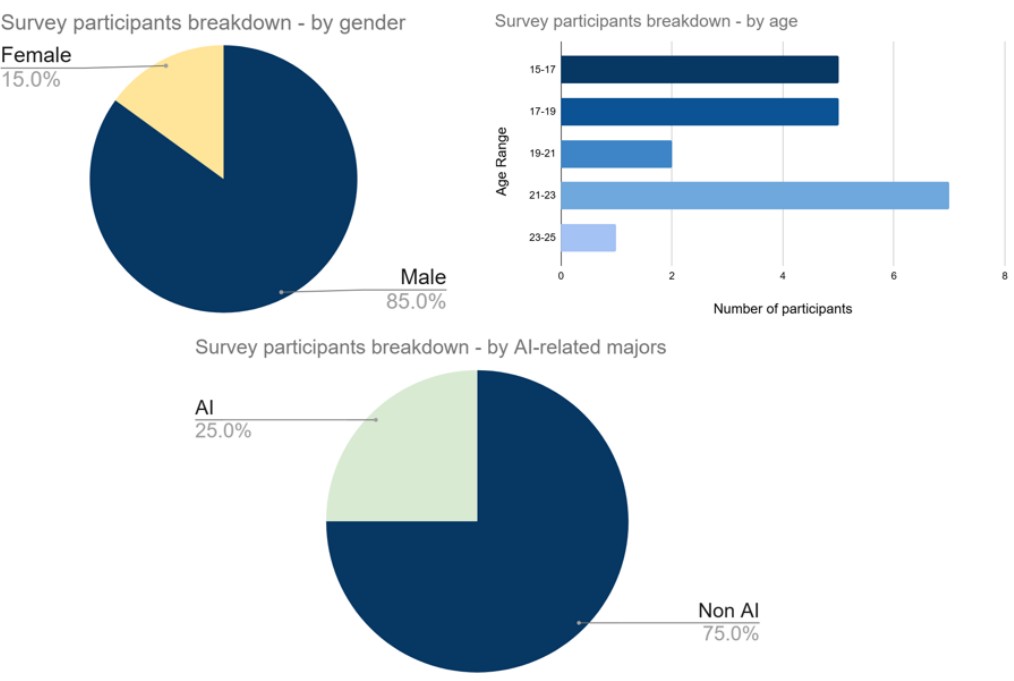

Figure 12: The information of users.

**Setup**  All methods were evaluated using the T2I SD model with publicly available checkpoints v1.5. We organized the participants into 20 batches, each randomly selecting 50 samples from a pool of 104 samples and shuffling the methods for evaluation. The original image and the images generated by the four methods were presented side-by-side for evaluation. To ensure objectivity, we blinded the method so that participants did not know which method the image belonged to, including our method. To ensure a fair comparison and achieve optimal results, we conducted our experiments and followed the recommendations provided by the authors. For samples that required retaining the object, we selected SDEdit with a strength of 0.5 and MasaCtrl with step 4 and layer 6. For other samples, we chose SDEdit with a strength of 0.8 and MasaCtrl with step 4 and layer 10.

**Tasks**  The participants were asked to rate the performance of each of the four methods on a scale of 1 to 6 for four metrics based on their perspectives. They must follow the order of samples and

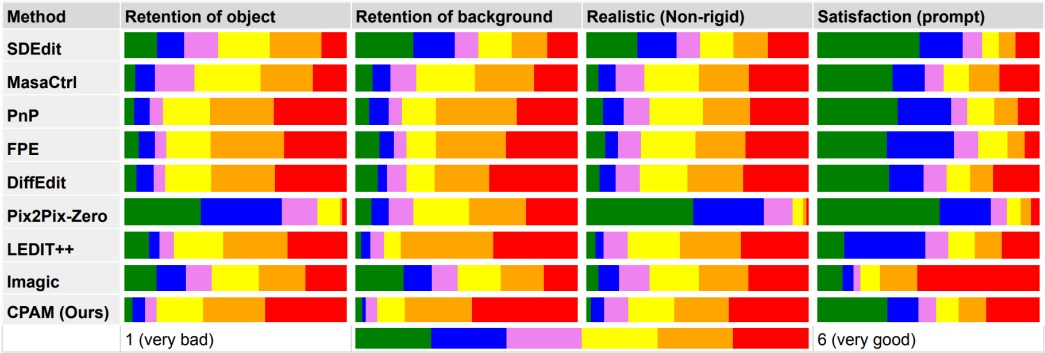

Figure 14: Without fine-tuning the model, it cannot generate novel views and poses of objects aligned with the text prompt. We compare these tuning-free methods to the fine-tuning method Imagic, which can generate novel views and poses.

methods in their batch. For samples where retaining the object was not required, participants left the rating blank in the cell corresponding to the retention of object metric.

**Apparatus and procedure** Our pilot study was conducted online and in our lab, where participants completed the assigned tasks in their respective batches. The total time for these study sessions was approximately 4 hours per person. Some sessions were video-recorded for further analysis.

**Quantitative results** We present the average rating scores of the participants, ranging from 1 to 6 (1 denoting "very bad" and 6 indicating "very good"), obtained from the user study evaluation. The results suggest that users expressed a high degree of contentment with CPAM in terms of object retention, background retention, and realistic metrics. Additionally, they rated high the satisfaction metric with the Imagic method (see Figure. 13).

| Method | Retention of object | Retention of background | Realistic (Non-rigid) | Satisfaction (prompt) |
|---|---|---|---|---|
| SDEdit | | | | |
| MasaCtrl | | | | |
| PnP | | | | |
| FPE | | | | |
| DiffEdit | | | | |
| Pix2Pix-Zero | | | | |
| LEDIT++ | | | | |
| Imagic | | | | |
| CPAM (Ours) | | | | |
| | 1 (very bad) | | | 6 (very good) |

Figure 13: The statistical ratings for each method.

## A.7 FAILURE CASES

We visualize failure cases as discussed in the limitations subsection regarding the limitations of pre-trained models 14, the instability of aggregated masks 15, editing specific spatial regions, and the model's focus on salient objects 16.

## A.8 FUTURE WORK AND FURTHER DISCUSSION

Our findings in using cross-attention to condition multiple text prompts for different regions offer a promising approach for editing images with multiple simple prompts. This stands in contrast to methods that rely on complex details and highly precise prompts for optimal results. In future work,

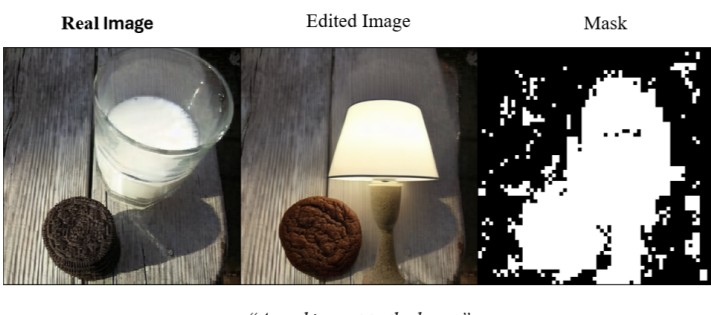

*"A cookie next to the lamp."*

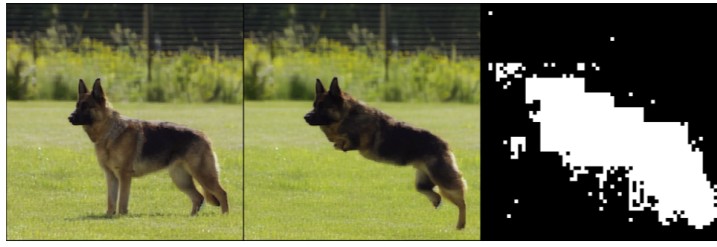

*"A photo of a jumping dog."*

Figure 15: The failure case occurs when the mask is imprecise. In the first sample, the mask often captures all the salient objects within the image and cannot focus solely on the desired object, even if we aggregate attention maps correlated to a specific token like "lamp". In the second sample, the dog's mask loses its legs.

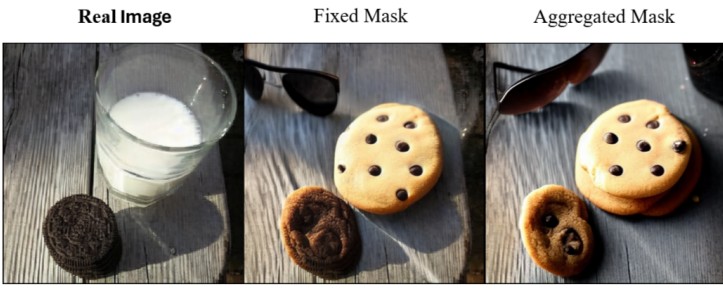

*"A cookie next to a sunglass."*

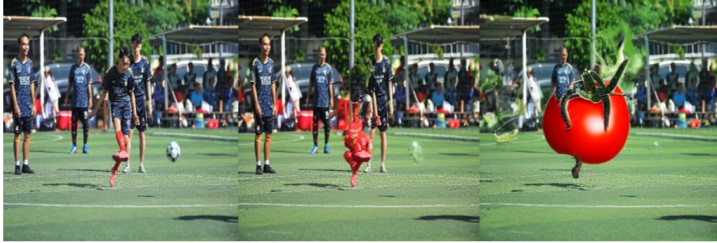

*"The player kicks the tomato."*

Figure 16: We modify the specific object within the image using fixed masks and aggregated masks as guidance. In the first sample, we aim to replace the glass of milk with a sunglass. However, the model generates the content in the wrong expected location. In the second sample, our goal is to replace the ball with a tomato. However, the model focuses solely on the salient object (people), resulting in an incorrect outcome.

we aim to leverage this technique for real-world image editing tasks, simplifying the user experience while maintaining high-quality outputs.

As our method relies on a precise mask, we plan to integrate a noise estimation technique Brack et al. (2024); Couairon et al. (2023) for generating masks, offering users a more robust solution. We can first generate an edited image to address the issues of the model generating content in the wrong place or being uncertain about where the model generates, especially in the task of editing a specific spatial region. Based on this, we can adjust the mask, prompt, and then re-generate the image. Addressing these problems constitutes our future work.

