# OpenReview forum: "Rethinking Attentions in Zero-Shot Real Image Editing"
_ICLR.cc/2025/Conference — Submitted to ICLR 2025_

### Official Review · Reviewer_3qqf · 2024-10-26

**Soundness:** 2
**Presentation:** 2
**Contribution:** 2
**Rating:** 5
**Confidence:** 3

**Summary:**

This paper proposes a Context-Preserving Adaptive Manipulation method for real image editing. Preservation adaptation module and localized extraction module are introduced for keeping objects maintained and mitigating interferences. Additionally, some mask-guidance strategies and a new benchmark are mentioned. The proposed method demonstrates better results in terms of context maintenance.

**Strengths:**

1. The insight of the paper is good, and the implementation of image editing with background preservation is close to the actual application scenarios.
2. The visualized results show that the proposed method achieves the desired effect.
3. The structure of the paper is well organized.

**Weaknesses:**

1. Although the problems attempted to be solved are interesting, the solutions lack innovation.
2. The additional introduction of mask inputs leads to unfair comparisons with existing methods.
3. The proposed mask-guidance strategies are rule-based and not flexible enough to cope with diverse editing needs.
4. Inadequate visual comparison of ablation experiments.
5. The structure and presentation of figure 2 is not sufficiently clear.
6. English expression needs improvement: “retaining object is true”, “where s, l, S = 3, L = 8 denotes” and “element-wise dot product”.

**Questions:**

1. How much does the accuracy of the input mask affect CPAM?
2. What will be the result of acquiring masks for CPAM inputs by means of adaptive perception or reference segmentation?

---

> ### Author Response · Authors · 2024-11-21
> **Response to Reviewer  3qqf (Part: 1/2)**
>
> ## **Weaknesses**
> ### **1. Innovation**
>
> We respectfully highlight the core contributions and novel aspects of our method:
>
> #### **A. Framework Innovation**
> - Our method integrates **Preservation Adaptation**, **Localized Extraction**, and a flexible **Mask-Guidance Strategy**, enabling diverse editing tasks such as rigid and non-rigid changes, object replacement, and multi-prompt edits with background consistency.
> - These advancements surpass baseline methods by offering precise, user-friendly, and adaptable solutions for real-world editing tasks.
>
> #### **B. Preservation Adaptation Module**
> - Independently manages object and background preservation across layers and steps, ensuring the background remains intact while providing precise control over object edits.
> - Enables tasks such as **object removal** and **background replacement** (see Figure 8 in Appendix A.2 for comparison with MasaCtrl).
>
> #### **C. Localized Extraction Module**
> - Introduces **multi-condition editing** using text and null-text prompts for precise, adaptive edits. This ensures non-target regions remain untouched while allowing seamless integration of multiple conditions.
>
> #### **D. Mask-Guidance Strategy**
> - Handles rigid and non-rigid edits within a unified framework, offering unparalleled flexibility. Simple mask adjustments allow tasks like **object replacement**, **background alteration**, and **object removal**.
>
> #### **E. Scalability and Usability**
> - **Multi-Prompt and Multi-Region Editing:**
>    Scales seamlessly to scenarios like "wear sunglasses" and "wear an Apple Watch," outperforming baseline methods limited to single-prompt, single-mask setups.
> - **Balancing Automation and Manual Control:**
>    Supports both user-drawn and automatic masks, ensuring adaptability for diverse user needs.
>
> #### **F. Advanced Insights into Attention Mechanisms**
> - **Cross-Attention:**
>    Enables precise conditioning for diverse edits while maintaining background integrity through null-text prompts.
> - **Self-Attention:**
>    Ensures cohesive edits and semantic consistency, enabling smooth transitions and precise exclusion for tasks like object removal.
>
> These contributions demonstrate meaningful advancements in flexibility, precision, and control, addressing limitations in existing methods and offering valuable innovations for real-world image editing. Thank you for the opportunity to clarify the novelty of our approach.
>
>
> ### **2. The additional introduction of mask inputs leads to unfair comparisons with existing methods.**
>
> We respectfully disagree with the claim that the use of masks leads to unfair comparisons. Mask usage is a widely accepted practice in image editing tasks, and many state-of-the-art methods also rely on masks. While some of the compared methods do not use masks, their objectives and constraints differ from ours, making their inclusion in comparisons still relevant.
>
> Additionally, we have compared our method with several mask-guidance approaches, such as **MasaCtrl**, **LEDIT++**, and **DiffEdit**. For these methods, we conducted a thorough and fair evaluation, rigorously searching for their best results using either their provided masks or masks generated by our approach. This ensures a comprehensive and unbiased comparison across all methods. We believe this evaluation framework demonstrates the practicality and effectiveness of our method in diverse scenarios.
>
> ---
>
> ### **3. The proposed mask-guidance strategies are rule-based and not flexible enough to cope with diverse editing needs.**
>
> We acknowledge that our mask-guidance strategy is rule-based, this was a deliberate design choice aimed at addressing a core challenge in image editing: achieving both rigid and non-rigid semantic transformations within a unified, adaptable framework.
>
> Our mask-guidance mechanism offers the following advantages:
> - **Precision and Clarity:**
>   It simplifies complex editing tasks by centralizing control through mask adjustments, eliminating the need for **architectural modifications or altering the flow pipeline**. This ensures flexibility and efficiency across diverse manipulation scenarios.
> - **Resolving Ambiguity:**
>   Consider the prompt, "a photo of a tree taken in autumn." Without mask guidance, it is unclear whether the goal is to alter the tree’s foliage, the background, or another element. Our approach resolves such ambiguities by allowing users to focus edits on specific regions.
> - **Wide Applicability:**
>   The rule-based strategy bridges a significant gap in image editing by enabling tasks ranging from subtle refinements to extensive transformations with minimal configuration changes.
>
> This combination of rule-based precision and user-driven flexibility makes our approach practical, scalable, and highly effective for real-world editing challenges.

---

> ### Author Response · Authors · 2024-11-21
> **Response to Reviewer 3qqf (Part: 2/2)**
>
> ### **4. Inadequate visual comparison of ablation experiments.**
>
> Thank you for pointing this out. We have expanded the visual comparisons for the ablation studies in the attached rebuttal pdf, highlighting the contribution of each module (e.g., Preservation Adaptation and Localized Extraction) to CPAM’s performance. These visuals demonstrate:
> - The effect of enabling/disabling each module on editing quality and background preservation.
> - CPAM delivers precise and localized edits while maintaining high realism across different mask choices.
>
> ---
>
> ### **5. The structure and presentation of Figure 2 are not sufficiently clear.**
>
>
> Thank you for your feedback. Figure 2 illustrates **multi-text guided synthesis**, where:
> - **Prompt 1 and Mask 1** (orange) condition one part of the latent noise.
> - **Prompt 2 and Mask 2** (purple) condition another part.
>
> The figure showcases CPAM’s ability to process multi-condition inputs, producing an image guided by both prompts. The caption explicitly describes this process and suggests zooming in for clarity. While we believe this effectively demonstrates our method’s capabilities, we are open to further adjustments based on specific suggestions.
>
> ---
>
> ### **6. English expression needs improvement.**
> Thank you very much. We appreciate your feedback and carefully proofread our manuscript for greater clarity and correctness.
>
> ## **Questions**
>
> ### **1. How much does the accuracy of the input mask affect CPAM?**
>
> The accuracy of the input mask influences CPAM's performance, but our framework is designed to be resilient to varying levels of mask precision. While precise masks enhance control and enable fine-grained edits, CPAM does not strictly require high-precision masks to function effectively. For instance:
> - **The fine-grained manual mask**, **the coarse manual mask** or **approximations** from off-the-shelf models can guide the editing process successfully.
> - Imperfect masks still yield meaningful edits, with users able to refine them interactively if desired.
>
> The results in our paper demonstrate CPAM’s robustness with diverse mask generation methods, as visualized in the attached rebuttal pdf.
>
> ---
>
> ### **2. What will be the result of acquiring masks for CPAM inputs by means of adaptive perception or reference segmentation?**
>
> Acquiring masks through **adaptive perception** or **reference segmentation** would significantly enhance the practicality and versatility of CPAM. Specifically:
> - **Automated Mask Generation:**
>   These methods enable automatic mask creation based on contextual understanding or predefined references, reducing the reliance on manual input.
> - **Practical Usability:**
>   This adaptability ensures CPAM can perform effectively in real-world scenarios where precise manual mask creation is impractical.
> - **Streamlined Editing Process:**
>   Automated masks make the editing process more accessible to non-expert users while maintaining flexibility for specific refinements.
>
> This capability would further position CPAM as a practical solution for diverse and complex image editing needs.

---

> ### Author Response · Authors · 2024-11-24
> **Looking forward to reviewer's reply**
>
> Dear Reviewer,
>
> Thank you for your valuable feedback during the review process. Our rebuttal carefully addressed your concerns raised in the initial review. As the deadline for the discussion phase approaches, we kindly remind you to read our rebuttal. If you have any questions, suggestions, or further clarification on any points, please feel free to reach out.
>
> We look forward to your feedback and hope for a positive outcome.
>
> Thank you for your time and consideration.
>
> Best regards,
>
> Authors of Paper 4584

---

> > ### Comment · Reviewer_3qqf · 2024-11-25
> > **Reply to rebuttal**
> >
> > Thanks to the authors' careful responses, my concerns about the visualisation of the ablation experiments and the presentation of the paper have been addressed. However, as with other reviewers, I still have major concerns about innovativeness, so I have chosen to keep my rating.

---

> > > ### Author Response · Authors · 2024-11-25
> > > **Response to Reviewer 3qqf**
> > >
> > > Dear Reviewer 3qqf,
> > >
> > > Thank you for your valuable response. We carefully address your concerns.
> > >
> > > >  I still have major concerns about innovativeness.
> > >
> > > *Your initial reviews did not mention that our work has any innovative issues.* However, we also would like to summarize our novelty in these proposed modules:
> > >
> > > - **Preservation Adaptation:**  An independent mechanism to manage object and background preservation across steps and layers, achieving precise edits without compromising non-target regions.
> > > - **Localized Extraction:** Uses both text and null text prompts for adaptive multi-condition editing.
> > > - **Mask-Guidance Strategy:** Effectively handles rigid and non-rigid changes for tasks like object removal and background replacement.
> > >
> > >
> > > By introducing these novel modules, our method significantly outperforms the baseline MasaCtrl. In addition, please see Appendix A.2 section in our paper for differences in our proposed architecture compared with MasaCtrl.
> > >
> > > We hope these clarifications resolve your concerns and are happy to address further questions to increase your rating.
> > >
> > > Thank you for your time and thoughtful review.
> > >
> > > Best regards,
> > >
> > > Authors of Paper 4584

---

### Official Review · Reviewer_RuP3 · 2024-11-02

**Soundness:** 2
**Presentation:** 2
**Contribution:** 1
**Rating:** 5
**Confidence:** 5

**Summary:**

This paper proposes Context-Preserving Adaptive Manipulation (CPAM) for zero-shot real image editing. It includes three main components: 1) a preservation adaptation module that adjusts self-attention mechanisms to effectively preserve and independently control the object and background; 2) a localized extraction module that applies attention to the spatial pixels of the extracted object from the feature query to the target prompt, while the remaining pixels attend to a null text prompt; and 3) various masking strategies tailored to different editing needs.

**Strengths:**

1. This paper is well-written and esay to read.
2. The performance of this paper is promising, achieving state-of-the-art results across various metrics.
3. The insights on multi-text guided synthesis (Fig. 2 and Sec. 3.4) are interesting.

**Weaknesses:**

1. Lacking of experimental evidence supporting the claim that "null text does not affect the output," as stated in Lines 206-211.
2. The notation "t > T, l > L" in Fig. 3(b) is inconsistent with "s > S, l > L" in Eq. 2, and Fig. 3 requires further refinement.
3. The proposed method, CPAM, seems incremental because: 1) the main design, "Preservation Adaptation," is highly similar to "Mask-Guided Mutual Self-Attention" from the MasaCtrl paper; and 2) the masking strategy, which aggregates cross-attention maps across all steps and layers, also closely resembles that of MasaCtrl in Sec. 4.2.
4. Lacking of ablation studies to validate the effectiveness of the proposed modules; only results under different classifier-free guidance scales are presented in Sec. A.1.

**Questions:**

I look forward to your response to address my concerns outlined in the 'Weaknesses' section. I will adjust my score based on your reply and the ratings from the other reviewers.

---

> ### Author Response · Authors · 2024-11-21
>
> ### **1. Lack of Experimental Evidence Supporting Null Text Claim**
> Thank you for your feedback but we respectfully disagree with the assertion that our claim lacks experimental evidence. The examples provided in **Figure 4** and **Figure 5** directly support our statement that "null text does not affect the output."
>
> - **Figure 4:**
>   In a real image editing scenario, the soccer ball is transformed into a rugby ball using the prompt "Messi and a rugby" with mask guidance. While MasaCtrl either modifies unintended regions or fails to solve the task, our **Localized Extraction (LE)** module preserves the background and precisely edits the soccer ball. This demonstrates that conditioning the background with null text ensures no unwanted changes occur.
>
>
> When synthesized from noise with null text, there is no noticeable change, as expected in unconditioned-guided synthesis. This behavior aligns with the inherent nature of null text guidance, which does not introduce modifications but maintains the original content.
>
> These examples validate our claim and highlight how null text guidance preserves content as intended.
>
> ---
>
> ### **2. Inconsistent Notation in Fig. 3(b) and Eq. 2**
> Thank you for pointing out this inconsistency in notation. We will update **Eq. 2** to ensure consistency with the notation used in Fig. 3(b) in the camera ready version.
>
> ---
>
> ### **3. Incremental Design Concerns Regarding CPAM**
>
> **Comment:**
> 1. "Preservation Adaptation" is similar to "Mask-Guided Mutual Self-Attention" from the MasaCtrl paper.
> 2. The masking strategy, which aggregates cross-attention maps across all steps and layers, also resembles that of MasaCtrl in Sec. 4.2.
>
> **Response:**
> While we acknowledge similarities between certain aspects of our method and MasaCtrl, we would like to clarify the distinctions and innovations:
>
> 1. **Preservation Adaptation vs. Mask-Guided Mutual Self-Attention (MasaCtrl):**
>    - While both methods manage object and background preservation, our **Preservation Adaptation Module (PA)** introduces significant differences:
>      - The **background** is consistently queried from the original image, ensuring its integrity throughout the editing process.
>      - The **object features** are managed independently across steps and layers, providing more precise control over object edits without unintended background modifications.
>    - These independent controls enable novel tasks, such as **object removal** and **background replacement**, which are challenging to achieve with MasaCtrl. Please refer to **Figure 8 in Section Appendix A.2** for detailed comparisons of architectural distinctions.
>
> 2. **Mask-Guidance Strategy:**
>    - Aggregating cross-attention maps for object mask generation is a common approach in the field, as seen in methods like **P2P**, **DiffuMask**, and **LEDIT++**. However, our method refines this process by:
>      - Providing multiple options for adjusting and selecting masks (manual, soft masks, off-the-shelf models, or automatic masks from our model).
>      - Supporting both **rigid** and **non-rigid edits** within a unified framework, which enhances flexibility and adaptability.
>    - This flexibility ensures CPAM can efficiently handle diverse editing tasks that baseline methods often struggle with, particularly in complex scenarios requiring nuanced mask-based control.
>
> 3. **Localized Extraction:**
>     - The Localized Extraction module is pioneering multi-condition editing by leveraging both text and null text prompts. This innovation ensures precise and adaptive control, enabling seamless integration of multiple conditions for complex editing tasks while maintaining the integrity of non-target regions.
>
> ---
>
> ### **4. Ablation Studies**
> We have conducted a detailed ablation study, included in the attached rebuttal pdf, to isolate and analyze the impact of each module in CPAM:
>
> - **Preservation Adaptation Module (PA):**
>   Disabling PA results in unintended changes to the background during edits, demonstrating its necessity for preserving non-target regions.
>
> - **Localized Extraction Module (LE):**
>   Disabling LE leads to interference in non-target regions, highlighting its role in ensuring precise, localized edits.
>
> These studies demonstrate the importance of both modules in achieving high-quality edits and underscore the novelty of our approach.

---

> ### Author Response · Authors · 2024-11-24
> **Looking forward to reviewer's reply**
>
> Dear Reviewer,
>
> Thank you for your valuable feedback during the review process. Our rebuttal carefully addressed your concerns raised in the initial review. As the deadline for the discussion phase approaches, we kindly remind you to read our rebuttal. If you have any questions, suggestions, or further clarification on any points, please feel free to reach out.
>
> We look forward to your feedback and hope for a positive outcome.
>
> Thank you for your time and consideration.
>
> Best regards,
>
> Authors of Paper 4584

---

> > ### Comment · Reviewer_RuP3 · 2024-11-25
> >
> > Thank you for the authors' detailed response. I sincerely appreciate your efforts in addressing the concerns. However, my primary concern about the novelty remains. The contribution appears to be an engineering extension of MasaCtrl. While the multi-text-guided synthesis is intriguing and achieves localized extraction, the overall contribution seems limited.

---

> > > ### Author Response · Authors · 2024-11-25
> > >
> > > Dear Reviewer RuP3,
> > >
> > > Thank you for your time and thoughtful review.
> > >
> > > > The contribution appears to be an engineering extension of MasaCtrl.
> > >
> > >
> > > Built up on MasaCtrl, we introduce novel modules which result in the outperformance of our method compared with the baseline MasaCtrl (Please see Appendix A.2 section in our paper for differences in our proposed architecture compared with MasaCtrl):
> > >
> > > - **Preservation Adaptation:**  An independent mechanism to manage object and background preservation across steps and layers, achieving precise edits without compromising non-target regions.
> > > - **Localized Extraction:** Uses both text and null text prompts for adaptive multi-condition editing.
> > > - **Mask-Guidance Strategy:** Effectively handles rigid and non-rigid changes for tasks like object removal and background replacement.
> > >
> > > We hope these clarifications resolve your concerns and are happy to address further questions.
> > >
> > > Best regards,
> > >
> > > Authors of Paper 4584

---

### Official Review · Reviewer_rvmA · 2024-11-04

**Soundness:** 2
**Presentation:** 2
**Contribution:** 2
**Rating:** 5
**Confidence:** 4

**Summary:**

The method proposes mask-guided image editing.

**Strengths:**

Editing performance is improved from baselines.

**Weaknesses:**

The method lacks novelty. The method of self-attention injection is already well-known method, and usage of mask-guidance is not a novel method. It seems the proposed method is combination of framework of Diffedit (automatic mask generation) and self-attention injection similar to Plug-and-play diffusion features. I think there is no new or technical idea which can further contribute to the generative AI field.

Also, the quality of output edited images are not satisfactory. The paper proposes that the editing method can be applied to all kinds of editing, but it seems that the outputs can be obtained with using baseline methods with proper parameter control.

**Questions:**

See Weakness

---

> ### Author Response · Authors · 2024-11-21
> **Response to Reviewer  rvmA (Part: 1/2)**
>
> ### **1. Novelty and Technical Contributions**
> #### **Preservation Adaptation Module**
> Thank you for your feedback. However, we respectfully disagree with the claim that our method lacks novelty or technical contribution. Below, we highlight the unique aspects of our method and its contributions to the field:
>
> - **Self-Attention Injection:**
>   While self-attention injection is a known technique, its implementation varies significantly across methods. For example:
>   - **Plug-and-Play (P2P)** swaps cross-attention maps.
>   - **FPE** swaps or injects self-attention maps.
>   - **MasaCtrl** replaces key-value (K, V) features without swapping maps.
>
>   Building upon MasaCtrl, our **Preservation Adaptation Module** introduces significant advancements:
>   - The **background** is consistently queried from the original image, ensuring it remains intact during editing.
>   - The **object** is meticulously controlled across layers and steps to achieve high guidance precision without distorting the background.
>
> This approach enables novel tasks, such as **object removal** and **background replacement**, which are difficult to achieve with existing methods. Figure 8 in Section Appendix A.2 provide detailed visualizations of architectural differences compared to MasaCtrl. Additionally, in the introduction of our paper, we address the limitations of existing methods leveraging self-attention or cross-attention, and we explain how our approach overcomes these shortcomings.
>
> #### **Localized Extraction Module**
>
> Unlike **DiffEdit**, which rigidly blends original and edited latent noise for local edits guided by a mask generated using noise estimation, our **Localized Extraction (LE)** module introduces a more flexible and adaptive approach:
>
> - **Multi-Conditioning with Text and Null-Text Prompts:**
>   By incorporating both text and null-text prompts, the LE module avoids interference in non-target regions, ensuring that edits are localized precisely to the intended areas.
> - **Adaptive and Non-Rigid Control:**
>   Instead of blending latent noise rigidly, the LE module offers greater flexibility and precision over the editing process. This approach supports nuanced and complex edits that adapt dynamically to the input.
>
> These advancements enable CPAM to deliver superior results, particularly in scenarios requiring precise localized edits or subtle adjustments, outperforming methods like DiffEdit in both accuracy and adaptability.
>
> #### **Mask-Guidance Strategy:**
>   Our mask-guidance strategy unifies **rigid** and **non-rigid** semantic changes within a single framework. By simply adjusting the mask, users can:
>   - Perform diverse image editing tasks efficiently.
>   - Achieve capabilities beyond what baseline methods can offer, enabling a broader range of flexible and user-friendly editing scenarios.
>
> ---
>
> ### **2. Scalability and Flexibility**
>
> Our method offers scalability and flexibility that baseline methods cannot achieve, especially in **multi-condition editing scenarios**. For example:
>
> - To edit an image with prompts such as **"wear sunglasses"** and **"wear an Apple Watch"**, users can decompose these into separate prompts and assign corresponding masks to the eye and wrist regions. This enables precise and independent edits for each region of the image. Baseline methods, which typically handle **single-prompt, single-mask scenarios**, are not equipped to handle such tasks seamlessly.
>
> - Our method supports flexible mask-guided editing, allowing users to:
>   - Preserve background integrity while editing specific objects.
>   - Combine multiple prompts with independent controls for each region, which significantly enhances the scope and applicability of image editing.
>
> These features demonstrate the unique scalability and flexibility of our approach, offering a distinct advantage over baseline methods.

---

> ### Author Response · Authors · 2024-11-21
> **Response to Reviewer  rvmA (Part: 2/2)**
>
> ### **3. High Guidance with Background Preservation**
>
> Our method excels in delivering **highly guided edits** without distorting the original background, a significant challenge for baseline methods. Key strengths include:
>
> - The **Preservation Adaptation Module:**
>   - Separately processes object and background information to ensure precise alignment with the text prompt while retaining non-edited regions intact.
>
> - The **Localized Extraction Module:**
>   - Avoids interference in non-target regions by using text and null-text conditioning.
>
> Unlike many baseline methods that often introduce distortions in unintended areas, CPAM maintains the original background with high fidelity.
>
> ---
> ### **4. Quality of Output Edited Images**
>
>
> While automated metrics like FID are valuable for evaluating image editing performance, they do not fully capture the real-world experience of edit quality as assessed by humans [1]. These metrics are influenced by various factors and provide only a partial perspective on performance. Therefore, we use these metrics as **supplementary tools** rather than as the sole basis for evaluating CPAM’s effectiveness. To further verify CPAM's performance, we conducted a **user study** (see Table 2) and included detailed visualizations.
>
> [1] Rethinking FID: Towards a Better Evaluation Metric for Image Generation, CVPR 2024.
>
>
>
> ### **Summary of Novel Contributions**
>
> 1. **Preservation Adaptation Module:**
>    - Enables novel tasks like **object removal** and **background replacement.**
>    - Separates object and background processing for precise, high-quality edits.
>
> 2. **Localized Extraction Module:**
>    - Introduces multi-conditioning to achieve precise and non-intrusive edits in non-target regions.
>
> 3. **Mask-Guidance Strategy:**
>    - Unifies rigid and non-rigid semantic changes in a single framework.
>    - Supports multi-mask scenarios for enhanced flexibility.
>
> 4. **Scalability and Flexibility:**
>    - Scales to multi-condition editing, outperforming baseline methods in handling complex tasks.
>
> These contributions highlight the technical and practical advances introduced by our method, addressing the limitations of existing approaches and expanding the capabilities of generative AI for image editing.

---

> ### Author Response · Authors · 2024-11-24
> **Looking forward to reviewer's reply**
>
> Dear Reviewer,
>
> Thank you for your valuable feedback during the review process. Our rebuttal carefully addressed your concerns raised in the initial review. As the deadline for the discussion phase approaches, we kindly remind you to read our rebuttal. If you have any questions, suggestions, or further clarification on any points, please feel free to reach out.
>
> We look forward to your feedback and hope for a positive outcome.
>
> Thank you for your time and consideration.
>
> Best regards,
>
> Authors of Paper 4584

---

> > ### Comment · Reviewer_rvmA · 2024-11-25
> >
> > Although the authors partially addressed my concerns with additional experiments, I still think the paper lack novelty and the other reviewers consistently claimed same concerns on novelty issues. I think the method is limted on simple marginal extension from baseline work of MasaCTRL.

---

> ### Author Response · Authors · 2024-11-25
>
> ## Response to Reviewer rvmA
>
> Dear Reviewer rvmA,
>
> Thank you for your thoughtful feedback.
>
> > The authors partially addressed my concerns
>
> We carefully address your concerns. Could you please let us know if you have questions or additional points that we need to clarify?
>
> > I still think the paper lack novelty. I think the method is limted on simple marginal extension from baseline work of MasaCTRL.
>
> We respectfully disagree with your comment that our work lacks novelty. We would like to summarize our novelty in  these proposed modules:
>
> - **Preservation Adaptation:**  An independent mechanism to manage object and background preservation across steps and layers, achieving precise edits without compromising non-target regions.
> - **Localized Extraction:** Uses both text and null text prompts for adaptive multi-condition editing.
> - **Mask-Guidance Strategy:** Effectively handles rigid and non-rigid changes for tasks like object removal and background replacement.
>
>
> By introducing these novel modules, our method significantly outperforms the baseline MasaCtrl. In addition, please see Appendix A.2 section in our paper for differences in our proposed architecture compared with MasaCtrl.
>
> We hope these clarifications resolve your concerns and are happy to address further questions.
>
> Thank you for your time and thoughtful review.
>
> Best regards,
>
> Authors of Paper 4584

---

### Official Review · Reviewer_zHLC · 2024-11-04

**Soundness:** 3
**Presentation:** 3
**Contribution:** 2
**Rating:** 5
**Confidence:** 4

**Summary:**

- This paper presents Context-Preserving Adaptive Manipulation (CPAM) to address complex, non-rigid image editing using a tuning-free, zero-shot approach.
- It utilizes both self-attention and cross-attention mechanisms in Stable Diffusion to enable intricate and text-guided image edits.
- It proposes the Preservation Adaptation Process to adjust self-attention layers to retain the identity, texture, and shape of the object being edited while the background remains unchanged.
- It also includes a Localized Extraction Module to avoid undesired changes to other image areas, and it enables the selective application of cross-attention.
- It proposes various strategies to control the editing scope based on the task, e.g. object removal, replacement, or background alteration, and it allows users to define which parts of the image are editable.
- The paper also introduces a new benchmark to evaluate real-image editing models and provides extensive qualitative and quantitative comparisons to show the effectiveness of the proposed method.

**Strengths:**

- This paper offers useful advances for real-image editing by removing the need for fine-tuning and allowing flexible, zero-shot editing.
- This paper is organized well and gives clear explanations of each new component and how they work together to improve image editing.
- The author provides extensive experiments to show that the method works better than existing methods.
- This paper also introduces a new benchmark dataset for more comprehensive evaluation.

**Weaknesses:**

- The paper proposes various new modules, but does not delve into ablation studies that isolate and analyze the impact of each component/module within the framework.
- The teaser figure shows unintended changes in the horse's view and texture, despite no prompt specifying these edits. This suggests some loss of subject-specific details. This also contradicts the authors' claim in Figure 14, where they state that without fine-tuning, the model cannot generate novel poses or views. It raises questions about the proposed method's control over subject attributes.
- Table 1 shows only marginal improvements in metrics that measure overall image quality and background. This makes the advantage unclear. Also, the paper should include some notes on each metric. For example, CLIP score could mean either prompt alignment (image-text) score or the subject-level image similarity (image-image) score.
- The dependency on high-quality masks can limit practicality in fully automated editing and in situations where generating precise masks is difficult.
- Some recent related works in the same domain:
	ProxEdit: Improving Tuning-Free Real Image Editing with Proximal Guidance
	EmuEdit: Precise Image Editing via Recognition and Generation Tasks

**Questions:**

Could the authors provide an ablation study to isolate and analyze the impact of each component? Also, could the authors clarify the limitations of pose/view manipulation in the current framework?

---

> ### Author Response · Authors · 2024-11-21
> **Response to Reviewer zHLC (Part: 1/2)**
>
> ### **1. Ablation Studies**
> We appreciate the Reviewer’s comments. As suggested, we have conducted further ablation studies to isolate and analyze the contributions of each module. The results are included in the attached rebuttal pdf file, showing the impact of each module, namely, Preservation Adaptation (PA) and Localized Extraction (LE), in the framework. For instance, enabling both PA and LE ensures effective editing localized to the mask regions, while disabling one or both results in varying degrees of control loss or unintended changes.
>
> ---
>
> ### **2. Regarding the Horse Teaser Figure**
>
> The unintended changes in the horse's view and texture, despite no prompt specifying these edits, arise from a trade-off between preserving original features and generating novel poses or views during image editing problem. This trade-off is controlled through the Preservation Adaptation module, which queries semantic content at specific layers and steps in the denoising process.
>
> - **Early Queries (e.g., Layers 1, 3, 5; Steps 1, 3, 5):** Too much original content is preserved, resulting in outputs that resemble the original subject too closely.
> - **Late Queries (e.g., Layer 13):** Excessive semantic alterations lead to outputs that deviate significantly, producing unintended changes and failing to align with the text prompt.
>
> To illustrate this, Section A in the attached rebuttal pdf demonstrates how the Preservation Adaptation module adjusts querying steps and layers, achieving the desired balance between maintaining original features and generating creative, text-guided edits.
>
> #### **Clarification on Novel Poses or Views**
>
> We thank the Reviewer for the comment regarding the statement, "they state that without fine-tuning, the model cannot generate novel poses or views." We will revise the sentence in the paper, which should be "**...without fine-tuning, the model cannot generate novel poses of or views of objects aligned with text prompt.**"
>
> For example, the pre-trained Stable Diffusion v1.5 model can generate novel pose views such as "jumping" or "spreading" for animals. However, it struggles with prompts like "kissing" for animals. This limitation arises because the pre-trained model, with frozen weights, cannot effectively generate new poses aligned with text prompt.
>
> ---
>
> ### **3. Marginal Metric Improvements in Table 1**
> We agree that additional clarity on metrics is necessary. The metrics used in Table 1 include:
>
> - **FID:** Measures the quality of edited images.
> - **CLIP Score:** Evaluates image-text prompt alignment.
> - **LPIPS (Background):** Assesses background detail preservation.
> - **Inception Score (IS):** Measures image diversity and realism.
>
> We acknowledge that CPAM exhibits only slight improvements in CLIP Score and LPIPS but the CLIP Score is slightly lower than that of fine-tuning Imagic, while LPIPS is comparable to that of LEDIT++ and DiffEdit. However, these automated metrics do not fully capture the real-world experience of edit quality as assessed by humans [1]. Therefore, we use them as supplementary tools rather than drawing definitive conclusions about CPAM’s effectiveness solely based on these metrics. To further verify CPAM's performance, we conducted a user study (Table 2) and included visualizations. The user study results show that CPAM excels in preserving object details, maintaining background consistency, and achieving high realism, which is reinforced by positive user feedback.
>
> [1]  Rethinking FID: Towards a Better Evaluation Metric for Image Generation, CVPR 2024

---

> ### Author Response · Authors · 2024-11-21
> **Response to Reviewer zHLC (Part: 2/2)**
>
> ### **4. Dependency on High-Quality Masks**
> We recognize that our approach may not fully address every scenario in automated image editing. However, our method does not rely on overly precise masks. The masks presented in our paper were manually drawn, which may give the impression of higher precision. Moreover, we have designed a flexible, mask-based solution that supports interactive editing, allowing users to obtain masks through various methods, including manual drawing, the fine-grained manual mask, the coarse manual mask, approximations from off-the-shelf models, or automatically derived masks from our model. This flexibility ensures practicality, as even with the most advanced editing methods, users often need to refine inputs—such as prompts or masks—to achieve results that align with their specific intentions. To provide further clarity, we have included a visualization of the different mask generation methods—manual drawing, soft masks, and off-the-shelf models—in the rebuttal pdf file.
>
> ---
>
> ### **5. Lack of Related Work Discussion**
> ProxEdit [WACV24] improves image inversion for accurate reconstruction with reduced time compared to null-text inversion, combining it with MasaCtrl for editing. However, it primarily demonstrates results on synthesized images and depends on source prompts with a strong correlation between source and target prompts for effective edits.
>
> EmuEdit [CVPR24] is an instruction-based editing method using multi-task learning but requires additional training to generalize to new tasks.
>
> There are numerous studies on image editing across various approaches, such as inpainting, instruction-based editing, and others. However, **our CPAM stands out as a text-guided editing method that operates in a zero-shot manner, eliminating the need for source prompts or strict prompt correlations**. With preservation adaptation, localized extraction, and flexible mask-guidance strategies, CPAM provides robust and precise results for real image editing tasks without fine-tuning or additional labeled data.

---

> ### Author Response · Authors · 2024-11-24
> **Looking forward to reviewer's reply**
>
> Dear Reviewer,
>
> Thank you for your valuable feedback during the review process. Our rebuttal carefully addressed your concerns raised in the initial review. As the deadline for the discussion phase approaches, we kindly remind you to read our rebuttal. If you have any questions, suggestions, or further clarification on any points, please feel free to reach out.
>
> We look forward to your feedback and hope for a positive outcome.
>
> Thank you for your time and consideration.
>
> Best regards,
>
> Authors of Paper 4584

---

### Author Response · Authors · 2024-11-21
**Response to Reviewers: Innovation, Novelty, and Core Contributions**

We appreciate the reviewers’ feedback and the opportunity to clarify the novelty and contributions of our proposed method. Below, we summarize the key aspects that highlight the novelty and practical impacts.

---

### **A. Core Contributions and Novelty**

**1. Innovative Framework**
Our paper introduces a novel framework that includes:
- **Preservation Adaptation:** Independently manages object and background preservation across steps and layers to achieve precise edits without compromising non-target regions. Please kindly refer to **Figure 8 in Appendix A.2** for comparisons with MasaCtrl.
- **Localized Extraction:** Pioneers multi-condition editing by leveraging both text and null text prompts to ensure adaptive control over diverse scenarios.
- **Mask-Guidance Strategy:** Handles rigid and non-rigid changes efficiently, enabling tasks like object removal, background replacement, while preserving background integrity, eliminating the need for **architectural modifications or altering the flow pipeline**.

Our framework offers precise, flexible, and user-friendly solutions for real-world image editing tasks.

---

### **B. Practical Significance of Mask-Guidance**

**1. Resolving Prompt Ambiguity:**
Masks clarify users' intent in ambiguous prompts, such as "a photo of a tree in autumn," by allowing users to specify whether to modify the tree, the background, or specific regions.

**2. Precision for Local Edits:**
Masks simplify isolating intricate regions (e.g., jewelry, textures) and was seamlessly integrated with user-drawn or automatically generated inputs.

**3. High Guidance Without Distortion:**
Our approach supports high guidance scales while preserving background integrity and avoiding distortions, enabling accurate and realistic edits.

---

### **C. Scalability and Usability**

**1. Multi-Prompt and Multi-Region Editing:**
Our method can be extended to complex scenarios where multiple prompts and masks are applied to distinct image regions. For example:
- To edit an image with the prompts **"wear sunglasses"** and **"wear an Apple Watch,"** users can assign corresponding masks to the eye and wrist regions. This functionality exceeds the capabilities of baseline methods limited to single-prompt and single-mask workflows.

**2. Balancing Automation and Manual Control:**
Our method supports both automatic and user-drawn masks, ensuring adaptability to diverse user needs while maintaining precision and flexibility.

---

### **D. Valuable Insights from Self and Cross Attention**

Our exploration of self- and cross-attention mechanisms provides a foundation for advancing future image editing techniques:
- **Cross-Attention:**
  By attending specific prompts to distinct parts of the latent noise, our CPAM ensures precise conditioning for edits. Null-text prompts maintain background integrity while focusing object modifications.
- **Self-Attention:**
  Enables smooth transitions and cohesive edits by connecting pixels to themselves, ensuring semantic consistency and precise exclusion of regions (e.g., for object removal).

These insights emphasize the control and flexibility of our method, paving the way for future innovations in image editing.

---

### Conclusion

Our work introduces meaningful advancements in precision, flexibility, and adaptability for image editing, addressing key limitations of existing methods. We believe our innovative framework, practical mask-guidance strategy, and valuable insights into attention mechanisms demonstrate CPAM's novelty and impact in advancing real-world image editing techniques. We thank the reviewers for their constructive feedback and the opportunity to showcase our contributions.

We have revised the paper to address missing information, correct English expressions, resolve inconsistencies, and improve clarity and professionalism throughout the text.


We also provide ablation studies and demonstrate the impact of input mask accuracy on CPAM performance in [the anonymous rebuttal website](https://anonymous.4open.science/r/Anonymize-8672/Rebuttal.pdf).

---

### Meta-Review · Area_Chair_SWu6 · 2024-12-17

**Metareview:**

This work presents a Context-Preserving Adaptive Manipulation method for image editing task to identity is maintained and background is undistorted during the editing process. Two major raised concerns are limited contribution and lack of the important ablation study. It consistently receives four borderline reject from all reviewers. While part of concerns are addressed via rebuttal, the major concern about
novelty still remains by at least explicitly mentioned by three reviewers. While there is some difference, AC agrees with all reviewers that the contribution of this work which bares a lot similarity with MasaCtrl is not big enough to meet the conference bar. Considering all the comments and discussions, a decision of reject is made. Authors are advised to continue improving the work based on reviews and resubmit elsewhere.

**Additional Comments On Reviewer Discussion:**

Two main concerns from reviewers are limited contribution and lack of the important ablation study. The latter one is sort of addressed by authors with ablation study provided for each proposed module. But for the first one, author's response did not convince reviewers, who all choose to keep their original score. Since the concern is pretty consistent among all reviewers, and AC also checked the prior work MasaCtrl and agrees that there are quite a few similarities, thus the final decision of reject is made.

---

### Decision · Program_Chairs · 2025-01-22

Reject